# Development of a Culturally Adapted Dietary Intervention to Reduce Alzheimer’s Disease Risk among Older Black Adults

**DOI:** 10.3390/ijerph20176705

**Published:** 2023-09-02

**Authors:** Ashley R. Shaw, Mickeal N. Key, Samantha Fikru, Saria Lofton, Debra K. Sullivan, Jannette Berkley-Patton, Crystal M. Glover, Jeffrey M. Burns, Eric D. Vidoni

**Affiliations:** 1Department of Neurology, University of Kansas Medical Center, Kansas City, KS 66160, USA; mkey@kumc.edu (M.N.K.); sfikru@kumc.edu (S.F.); jburns2@kumc.edu (J.M.B.); evidoni@kumc.edu (E.D.V.); 2College of Nursing, University of Illinois Chicago, Chicago, IL 60612, USA; slofto4@uic.edu; 3Department of Dietetics and Nutrition, University of Kansas Medical Center, Kansas City, KS 66160, USA; dsulliva@kumc.edu; 4Department of Biomedical and Health Informatics, University of Missouri-Kansas City, Kansas City, MO 64108, USA; berkleypattonj@umkc.edu; 5Department of Psychiatry and Behavioral Sciences, Division of Behavioral Sciences, Rush Medical College, Chicago, IL 60612, USA; crystal_glover@rush.edu; 6Department of Neurological Sciences, Rush Medical College, Chicago, IL 60612, USA; 7Rush Alzheimer’s Disease Center, Rush University Medical Center, Chicago, IL 60612, USA

**Keywords:** diet intervention, older adults, black adults, community engagement, qualitative research, lifestyle interventions, Alzheimer’s disease

## Abstract

The objective of this study is to identify and understand knowledge and attitudes that influence dietary practices among older Black adults using a community-engaged approach. This is a non-interventional mixed methods study designed to inform the development of an adapted brain-healthy soul food diet intervention. A purposive sampling approach was used to conduct seven semi-structured focus group discussions and an online quantitative survey. In total, 39 participants who self-identified as Black, aged 55 years and older, English speaking, and who were cognitively normal with an AD8 < 2; (25.6% men; 74.4% women) participated in the online survey and one of the seven 60 min virtual focus group discussions (5–7 per focus group). Quantitative frequency data from the online surveys were analyzed using descriptive statistics. Qualitative focus group data were analyzed using a 6-step thematic analysis process. Five themes emerged: dementia awareness; practices shaping food choices and consumption; barriers to eating healthy; instrumental support; and elements of a culturally adapted brain-healthy dietary intervention. Older Black adults perceived an adapted MIND dietary model as the most acceptable with the incorporation of salient cultural characteristics and strategies within both the design and delivery phases.

## 1. Introduction

Black adults are two to three times more likely to develop Alzheimer’s disease (AD) compared to non-Hispanic Whites, with AD being the fourth leading cause of death among Black adults [1]. Previous research has indicated that healthy eating habits and requisite diets such as the Mediterranean diet (MeDi), Dietary Approaches to Stop Hypertension (DASH), and Mediterranean-DASH Intervention for Neurogenerative Delay (MIND) help reduce AD risk and slow cognitive decline among adults 65 years and older [2,3,4,5]. Large studies such as a 1.5 year study including non-demented older adults found that a higher adherence to the MeDi was associated with up to a 40% reduced risk for AD [3]. Another study found that adherence to the MIND diet, which is a combination of the MeDi and DASH diet, was associated with slower cognitive decline, with a rate reduction equivalent to 7.5 years younger and individual cognitive domains increased by 30–78% [2]. The MIND diet specifically incorporates foods from the MeDi and DASH that have been associated with a reduced risk of AD [6], slower cognitive decline [6], and reduced neuropathological changes of AD [7,8]. The MIND diet focuses on the consumption of plant-based foods (i.e., green leafy vegetable), nuts/seeds, berries, fish, and the use of extra virgin olive oil [9]. Additionally, the MIND diet limits consumption of red and process meats, margarine, cheese, fried foods, and sweets [9]. The mechanisms for the effect of the MIND diet on AD has been linked to a high consumption of omega 3 fatty acids, which led to a reduction in amyloid-β formation, oxidative stress, and inflammation [10,11]. Additionally, research has suggested that flavonoid metabolites may indirectly impact brain function and cognitive performance by modulating neuronal receptors and neurotrophins resulting in changes in the synaptic function [12]. The amyloid-β formation, oxidative stress, and inflammatory issues related to AD were improved following the MIND diet due to the consumption of green leafy vegetables and other vegetables that contain high levels of flavonoids, beta carotene, and carotenoids [5]. However, much of the research regarding the dietary prevention of AD is based almost exclusively on studies involving non-Hispanic White adults, thus limiting the generalizability of the findings for older adults who belong to racial/ethnic groups who remain underrepresented and understudied in aging research [2]. For studies that have included diverse cohorts, adherence to diet interventions is lower for Black adults [13]. Research postulates that this low adherence and acceptability is due to the perceived lack of social support, social contexts (i.e., cost of healthy foods; reduced access to healthy and diverse food options due to lower socioeconomic status; etc.), strong cultural influence on food preferences and preparation, and perceived less appealing taste of low fat foods [13,14,15]. Previous research has indicated that modifying traditional soul food to better meet nutritional guidelines would be more effective than suggesting that such foods do not belong in and must be eliminated from a healthy diet altogether [16]. Traditional soul food is an integral part of Black culture that embodies heritage, legacy, and evokes a strong sense of togetherness within the Black community [17]. Soul food is traditionally known for being rich in flavor and consisting of fried foods, processed and organ meats, animal fats, sodium, and added sugar [17,18]. It is important to note that soul food incorporates nutrient-dense foods like leafy green vegetables as seen within the MIND diet. Notably, traditional soul foods include healthy dietary components [19], such as fruits and vegetables (i.e., collard greens, sweet potatoes, okra, and blueberries), which are linked to improved cardiovascular risk markers and slower cognitive decline [20,21]. Prior work has demonstrated success in culturally adapting diets for Black adults, especially when such dietary interventions are guided by community participation [22]. For example, culturally adapted diets have resulted in lower body mass index (BMI), decreased hemoglobin A1c (HbA1c) levels, and reductions in body weight and waist circumference among Black adults [23,24]. Nutrition interventions that have incorporated cultural tailoring such as spirituality and the inclusion of traditional cultural foods have shown promise in weight loss and dietary outcomes among Black adults [25,26]. For example, in a randomized 14-week faith-based weight loss intervention, on average, Black women lost a total of 10 pounds and had a 2.5-inch decrease in waist circumference [26]. Additionally, interventions focused on cooking instruction have been shown to be successful among Black adults [27]. Kramer et al. (2023) found that Black adults who were randomized in a cooking intervention resulted in greater intentions to cook and increased healthy cooking behavior [27].

While the use of culturally tailored diets has seen positive outcomes for health issues related to body composition and glucose regulation, researchers have insufficiently explored diets specified for improvements in brain health and cognitive performance. The MIND diet appears to be the sole dietary approach specifically created to address cognitive decline in older adults [9], yet it has not yet been culturally tailored for Black communities. Adapting the MIND diet to center and include Black cultural food traditions first requires a foundational understanding of the needs and expectations of the community, gathered via community-engaged mixed-methods research.

Qualitative research is the “systematic collection, organization, and interpretation of textual material derived from talk or conversation. It is used in the exploration of meanings of social phenomena as experienced by individuals themselves, in their natural context” (Malterud, 2001, p. 483) [28]. In health disparities research, qualitative methods are uniquely suited to generate in depth and detailed descriptions by exploring cultural phenomena, attitudes, beliefs, motivations, and experiences [29]. In this study, qualitative research centers on participant perspectives and examines their experiences and beliefs as they relate to food and eating patterns. Qualitative research methods have been used to advance our knowledge and understanding regarding aspects of food and eating practices [30]. Focusing on attitudes, beliefs, practices, and the cultural meanings of health and dietary patterns from the community perspective can provide valuable context for future culturally appropriate dietary intervention strategies, as well as inform needed policy approaches [31]. Qualitative sampling rests on the notion of saturation, rather than generalizability that is seen in quantitative statistically driven research. Specifically, the qualitative research findings are associated with the emergence of themes derived from a thorough understanding of experiences. To strengthen the findings, the use of internal validity is used to evaluate rigor in qualitive research [29,32]. Contrarily, quantitative research methods focus on quantification, which allows for a more robust analysis of the data to enhance the generalizability of the research findings due to larger samples sizes that are randomly selected [33]. It is important to note that both qualitative and quantitative research methods have unique advantages that can extend to enhance our understanding of research questions.

Mixed-methods research (MMR) integrates components of both qualitative and quantitative research methods to better understand a phenomenon [34] and is well recognized in health-related research [35]. The use of MMR allows for the exploration of a more divergent outlook and provides contextual understanding shaped via cultural and real-world experiences [36,37,38]. Additionally, MMR allows for triangulation that enhances both the validity and credibility of the research findings [39]. Qualitative data coupled with the quantitative survey results allows for the confirmation and expansion of the qualitative findings, thus providing an enhanced foundation for developing a future intervention that is measurably accessible to the target population.

Recently, a growing body of literature has described the importance of MMR for community engaged research (CER) interventions as it relates to health and prevention [40,41]. CER is an approach that consciously involves community members impacted by a phenomenon incorporating their insights throughout the research process, which can support the development of interventions and influence policies [42,43]. CER has been widely recognized in research as an effective tool to reduce health disparities [44,45,46]. The use of CER within qualitative research can enhance the translational results. Previous research has found that studies incorporating a CER approach had a positive impact on health behaviors including diet, exercise, and health outcomes (i.e., quality of life) [47].

MMR and CER work synergistically to provide a comprehensive understanding to research questions and practical problems through the use of qualitative and qualitive data [48]. Incorporating both MMR and CER allow for a more contextually relevant and effective plan for health interventions. In order for dietary interventions aimed at enhancing brain health among older Black adults to be effective, cultural relevant factors need to be examined and understood from the community’s perspective, which can be captured using both CER and MMR to inform the foundational aspects of a brain-healthy diet intervention. Therefore, the objective of this study was to inform the dietary approaches to preventing cognitive decline among Black adults by identifying and understanding the knowledge and attitudes that influence their dietary practices using a mixed-methods approach.

## 2. Materials and Methods

### 2.1. Design

This was a non-interventional mixed-methods study conducted to inform the development of an adapted brain-healthy soul food diet intervention. Specifically, this study aimed to understand older Black adults’ specific food preferences, preparation preferences, and community cooking class topics specific to supporting the development and delivery of the future-adapted brain-healthy diet intervention. Approval to conduct the study was obtained by the University of Kansas Medical Center Institutional Review Board (IRB#00146894).

### 2.2. Participants and Recruitment

We used purposive and convenience sampling for recruitment. Purposive sampling focuses on selecting participants based on specific characteristics relevant to the purposes of the study while convenience sampling involves selecting participants that are conveniently accessible [49,50]. For this study, we used both purposive and convenience sampling to recruit participants from the Kansas City metropolitan area from March 2021 to May 2021. Specifically, we used existing outreach and recruitment registries, the distribution of flyers, and an email list server which was developed from previous outreach efforts of the principal investigator (ARS). Individuals interested in participating in the study contacted the research staff by phone and completed a brief telephone screening to verify their eligibility. Study eligibility requirements included self-identifying as Black or African American, 55 years and older, English speaking, and cognitively normal with an eight-item informant interview to differentiate an aging and dementia (AD8) screening of <2. AD8 was selected due to its sensitivity (>84%) in detecting early cognitive changes associated with dementia-related illnesses such as AD, in which a score of less than 2 indicates normal cognition [51].

### 2.3. Development of the Focus Group Guide

The Health Belief Model (HBM) provided a theoretical framework to explore perceptions of nutrition and brain health. The HBM was chosen because it has been widely used in diverse cultural context areas and within dietary-related intervention development [52,53,54]. Constructs of the HBM, including perceived susceptibility, perceived severity, perceived benefits, perceived barriers, self-efficacy, and cue to action [55], were the focus of interest to guide the development of the focus group discussion guide. The focus group discussion guide questions were based on the following constructs: perceived susceptibility of dietary habits and dementia, perceived benefits of healthy eating, perceived severity of poor nutritional consumption, perceived barriers to eating healthy, and cues to action for following a brain-healthy diet. The final focus group guide consisted of 17 questions.

### 2.4. Procedure

All participants provided informed, written consent before participating in the study. Prior to the focus group discussions, 39 participants completed a 30 min online quantitative survey consisting of demographic questions (15-item), eating habits questions (15-item), and needs assessment questions (20-item). Specifically, surveys included questions about demographics, health care, eating patterns, cultural food needs, and community nutrition education needs. The survey was administered via a REDCap online link that was emailed individually to each participant 7 days prior to their scheduled focus group discussion.

Seven focus group discussions, each consisting of 5–7 people (39 total), were conducted between May and July 2021. Our final sample size was consistent with prior guidance on achieving saturation in qualitative research, which is holding 4 to 8 focus group discussions with 6 to 12 participants in each group [56,57]. Each focus group took place virtually using the Zoom platform, and each focus group lasted 60 min. Focus groups followed a discussion guide in which open-ended, clear, and concise questions such as “Why do some people in the community not follow healthy eating practices?”, “What kind of support would the community need to help overcome challenges?”, and “What would a brain-healthy diet look like for the Black community?” were used to stimulate discussion. Additionally, probe questions such as “Could you tell me more about why some people in the Black community do not follow healthy eating practices?” and “Could you tell me how a brain-healthy diet intervention should be delivered to the community?” were used to obtain more detailed information from the focus group participants. All focus group discussions were audio recorded and transcribed verbatim. The focus groups were led by a trained and racially congruent moderator. A racially congruent moderator was chosen because previous research suggests that homogeneity between participants and a moderator enhances the participants’ comfortability, which leads to a more open and authentic discussion [58,59]. Participants received a $50 gift card in appreciation for their time.

### 2.5. Data Analyses

First and senior authors, ARS and EDV, led the analysis of both the quantitative and qualitative data. We used a convergent parallel design in which the qualitative and quantitative data were collected and analyzed simultaneously, prioritized equally, and kept separate until the interpretation process. The qualitative and quantitative findings were analyzed in parallel before the consideration of convergence.

For quantitative data analyses, we conducted basic descriptive analyses, particularly frequencies, using SPSS (Version 27.0. IBM Corp: Armonk, NY, USA). Online survey responses were calculated as a frequency of endorsement. All available data were included in the analyses. The missing data was sparse and considered to be randomly distributed, with complete data for more than 99% of questions. Transcripts of qualitative focus group discussions were uploaded into Dedoose [60] for coding, filing, and organizing the themes and subthemes.

Qualitative data was analyzed using an iterative and reflective Braun and Clarke (2006) 6-step thematic analysis process that involved continual movement back and forth between each of the phases as a means to establish trustworthiness in qualitative research [61]. The 6-step thematic analysis included (1) familiarization—the transcription and prolonged engagement of reading data and the documentation of thoughts about potential themes (2) initial coding—coding data in a systematic manner to identify preliminary codes, (3) generating themes—collecting codes and gathering relevant data for potential themes, (4) reviewing themes—research triangulation and a deeper review of identified themes that are vetted, (5) defining and naming themes—refining themes and potential subthemes, and (6) interpretation and reporting—member checking and organizing themes in a manner to produce a report that portrays the participants’ perception [62]. Specifically, during step 6 of the thematic analysis, we conducted member checking, or participant validation, to ensure the participants’ perspectives were appropriately represented and not curtailed by the researchers knowledge and personal agenda [63]. More specifically, we conducted a separate focus group to ensure that the emerging themes were congruent with the community’s interpretation, resulting in a final thematic consensus. The member check focus group was comprised of a subset of participants (*n* = 5), was led by the same moderator who led the seven focus group discussions, and it took place 3 months after the final focus group discussion via the Zoom platform. Participants in the member check focus group were randomly selected from a subset of participants who expressed a willingness to participate in future research-related activities. The member check focus group lasted 30 min and encouraged participants to check for data accuracy, which was consistent with best practices for exploring credibility and enhancing the rigor of qualitative methods [63,64].

## 3. Results

A total of 39 people participated in the study. Most participants were female (74.4%), 65 years and older (60.3%), with an education level equivalent or higher to a bachelor’s degree (61.5%), retired (69.2%), and who identified as Christian (94.9%). Additionally, most participants reported having cardiovascular-related health conditions such as hypertension (66.7%) or high cholesterol (51.3%). Additional demographic details about the sample are shown in Table 1.

### Online Survey Results

Eating Habits: Most participants indicated that they currently either do not follow a dietary eating practice (48.7%) or follow a low-sodium eating diet (25.6%). For food preparation, most participants reported regularly baking (82.1%), grilling (46.2%), or air frying (43.6%) their food. A full description of reported eating habits can be found in Table 2.

Health Conditions Knowledge Interest: Many participants indicated that they were “very interested” or “interested” in receiving more health education related to AD (89.7%) and blood pressure (76.9%). A majority of participants were also “very interested” or interested” in other cardiovascular health conditions including cholesterol (69.2%), diabetes (74.4%), and obesity (69.2%) (Table 3).

Barriers to Eating Healthy: Regarding the perceived barriers to eating healthy, many participants indicated other reasons (i.e., foods upsetting one’s stomach, food spoilage, lack of information on benefits, and sleep/travel schedule) (41%) and taste (30.8%) as primary barriers, followed by cost (20.5%). A full description of perceived barriers to eating healthy can be found in Table 4.

Nutrition Education and Delivery Preferences: For a dietary intervention, participants reported that they were either “very interested” or “interested” in receiving education related to water intake (94.9%), meal planning and prepping (87.2%), nutritional content (87.1%), and eating healthy on a budget (84.6%). Most people indicated that they preferred support from a nutritionist or health coach virtually (94.9%) or by phone (94.9%). See Table 5 for nutrition education and delivery preferences.

Focus Group Discussion Themes: A total of five themes emerged from the focus group discussions. The results are summarized according to the major themes and subthemes expressed across all seven focus group discussions. See Table 6 for a summary of themes, subthemes, and sample quotes.

Theme 1. Dementia Awareness

All participants reported various ways of first learning about dementia. For theme 1, we identified three subthemes: Family History, Perceived Cause of Dementia, and Role of Diet in Dementia.

Subtheme 1—Family History: Many participants expressed that they first learned about dementia from having a family history of dementia (i.e., grandparents, parents, in-laws, older siblings). “*I first learned about it with my father. My father was diagnosed with it and I just say if you’ve ever been diagnosed with dementia, might as well say you have Alzheimer’s* [sic]. *So that’s how I first learned about it; that was about 16 or 17 years ago.*” Some participants also mentioned that they first learned about dementia via media outlets (i.e., articles, American Association of Retired Persons) and through church, in which older congregant members at their church had dementia.

Subtheme 2—Perceived Cause of Dementia: Participants across all focus groups perceived dementia to be prevalent among older people in general, not necessarily more prevalent among Black adults compared to other racial/ethnic groups. However, several participants believed that dementia is serious in the Black community due to the lack of awareness the community has about what dementia is and the perception that it is a normal part of aging (i.e., senility). Participants voiced that they believed contributing factors to the lack of awareness stemmed from a lack of education and a lack of access to available resources. “*I’m feeling like it is a serious problem for Black people…. we don’t have the access or the resources or sometimes the will to go and find out what is going on with us or with a relative. On the other hand, I would hesitate to say definitively that this more serious for Black people ‘cause I have not seen statistics on that.*” This sentiment was consistent with the overwhelming endorsement of interest in further education on dementia (89.7%).

Subtheme 3—Role of Diet in Dementia: Most participants believed that following a healthy diet was important for keeping the brain healthy. “*I think that a healthy diet is important for every part of your body, especially the brain, because if we give the brain the correct nutrients, I just believe that we will, you know, function better and function longer if we eat the right things.*” Specifically, participants perceived that regularly following a healthy diet, especially early in life, can provide essential nutrients to keep the brain functioning healthy as we age. However, some participants voiced that they believed that healthy eating is not commonly practiced until one becomes older, when people start thinking more about longevity of life.

Theme 2. Practices Shaping Food Choices and Consumption

For theme 2, we identified three subthemes: Household Decision Making, Food Purchasing Practices, and Food Preparation Practices.

Subtheme 1—Household Decision Making: Many participants indicated that they lived alone; therefore, household decision making about food was made individually. Among those who indicated that they were married, many indicated that the wife performs a majority of the decision making regarding food. Lastly, the key factor that played a role within decision making for food was cost.

Subtheme 2—Food Purchasing Practices: Participants voiced that food purchasing practices were largely influenced by cost and that they shopped based on deals from grocery ads and the deals in their supermarkets. Some participants indicated that due to the COVID-19 pandemic, their food purchasing practices had shifted to primarily grocery shopping online, with the exception of going in-person to a grocery store to purchase produce. “*Since the pandemic, I have had like, uh, most of my shopping online and I go and pick it up, and I have a few smaller stores that I go to purchase things that, um, things like vegetables, things that I usually go in person to pick those out, not at a smaller store.*” Participants indicated that food purchasing practices emphasized variety in meals, including multiple trips to different stores.

Subtheme 3—Food Preparation Practices: Most participants voiced that they liked to use a variety of food preparation practices including baking, air frying, and grilling. It is important to note that many participants stated that frying foods was rarely practiced within their households. “*I sauté. I’m using the oven, as the weather continues to change, and now that I can make a good fire, I might use the grill more. But frying is just not something I do not do at all.*”

Theme 3. Barriers to Eating Healthy

All participants across focus groups reported various barriers that impacted their ability to practice healthy eating. For Theme 3, we identified four subthemes: Access, Cost, Taste, and Food Spoilage.

Subtheme 1—Access: Many participants voiced access to healthy foods as being a significant barrier to eating healthy in the Black community. Specifically, many participants indicated that there are food deserts in many of the predominantly Black neighborhoods coupled with poor public transportation, making it difficult to access healthy foods. “*In many of the communities they don’t have access to fresh fruits and vegetables…Places like when I’m walking is a food desert, you know they don’t have, uh, fresh produce in the stores in the neighborhoods, so if they don’t have it there, they can’t get it.*” Additionally, many participants stated that stores in predominantly Black neighborhoods do not offer fresh produce, resulting in community members purchasing processed foods that are readily available, which often are not healthy options.

Subtheme 2—Cost: A majority of the participants across all focus groups voiced that the cost of healthy food (i.e., fruits and vegetables) was a significant barrier to healthy eating in the Black community and believed that people would eat healthier if they could afford it. “*Cost. I think that, I think that if you eat healthier, it’s going to cost you more, instead of being an investment in your life, so some of the plant-based foods are more expensive, and some of the plant-based cheese are more expensive.*” Additionally, some participants believed cost was more of a barrier for people living in households with more than one generation. Overall, the cost of food inhibits the Black community from being capable of fully investing in their health due to the lack of affordability. Cost was the second most frequently noted barrier (20.5%) to healthy eating in the survey. Additionally, participants almost universally endorsed interest in learning to eat healthy on a budget (84.6%).

Subtheme 3—Taste: Some participants perceived taste as a barrier to consuming healthy food, noting that healthy food often tastes bland. Also, it was perceived that people in the Black community are more accustomed to foods that have higher sodium for seasoning purposes, which makes it less appetizing when consuming healthy foods of lower sodium. “*Sometimes people are used to the salt, and they don’t want to eat bland food.*” Indeed, a plurality (30.8%) of participants identified taste as a potential barrier to healthy eating on the survey

Subtheme 4—Food Spoilage: Participants indicated that they perceived healthy foods, particularly produce, to spoil quickly, hindering their ability to follow a healthy diet. “*I like vegetables, so it’s not a problem. The only problem is just keeping the vegetables on hand, you know without them spoiling and I have to throw them out.*” Specifically, people perceived fresh produce to have a short shelf-life, meaning that they would need to make multiple trips to the grocery store throughout the week, which can be burdensome.

Theme 4. Instrumental Support

For theme 4, we identified two subthemes: Cooking Education and Accessibility Guidance. The most important support mechanisms that were voiced across all focus groups to help people follow a healthy diet include education and accessibility guidance.

Subtheme 1—Cooking Education: Across all focus groups, participants highlighted the importance of providing cooking education to support buy-in from the community. “*I just think, more education. If you go out in the community and maybe even do demonstrations where you can show people that, how the food tastes, how easy it is to prepare.*” Many participants believed that cooking education should center around cooking demonstrations as a means to explore new foods. Cooking demonstrations, coupled with sample tasting, would allow the community to obtain a better sense of healthy food preferences that could be incorporated in their diet moving forward. Endorsement of classes on meal planning and prepping (87.2%) and nutritional content (87.1%) on the surveys was consistent with this subtheme.

Subtheme 2—Accessibility Guidance: Many participants indicated that although providing education (i.e., recipes, cooking demonstrations) is helpful in providing support to a healthier diet in the Black community, where it is imperative to show where and how people can obtain healthy foods (i.e., highlighting specific grocery stores, farmers markets). “*It also has to be available, we can educate, we can show, we can even provide taste, but we have to make it available and show them where they can obtain or acquire these healthy foods, these recipes, these meals.*”

Theme 5. Elements of Culturally Adapted Brain-Healthy Dietary Intervention

All focus groups discussed ideas about the design of a brain-healthy dietary intervention with the goal of improving healthy eating in the Black community to reduce the risk of AD. For theme 5, we identified four subthemes: Recruitment, MIND Dietary Model, Comprehensive Education and Delivery Method, and Retention.

Subtheme 1—Recruitment: Several participants stated the recruitment into a brain-healthy diet intervention needs to be comprehensive, meaning that it should include diverse outlets to reach a wide range of diverse older adults (across the metropolitan area) within the Black community. Specifically, recruitment via media outlets such as newspapers, radio, and TV ads would be best in reaching older Black adults. “*It needs to be put out comprehensively. Television, radio, magazines, newspapers, social media, churches, groups go into nursing homes and things of that nature; that’s, you know, a comprehensive approach because you can’t just utilize one level of approach. Because that’s only going to affect only a segment of that community.*” Many participants across focus groups also voiced the importance of establishing partnerships with local churches and community centers who could support recruitment efforts by distributing recruitment materials via newsletter and email. Notably, social media outlets (i.e., Facebook and Twitter) were not favorably viewed by many of the participants as a way to recruit older adults due to irregular use of the platforms.

Subtheme 2—MIND Dietary Model: Of the three diets (MIND, MeDi, and DASH) discussed at each of the focus group sessions, the MIND diet model was most favored and seen as the most feasible to follow due to the variety of food options and because it was viewed as aligning more with the lifestyle practices within the Black community. “*I like the variety; you know most of it, I’ve kind of been doing already. Not much of the olive oil, like I said I stick with kind of canola oil. So, it limits you on the cheese, I see that. I mean I use way more cheese than one serving a week. Everything else is OK.*” Moreover, the MIND diet was viewed as less restrictive compared to the MeDi. However, many participants voiced that some adjustments in the model are needed to increase acceptability within the Black community including increasing the recommended amount of butter and cheese, as well as providing plant-based alternative options in these categories. “*Less than one,…if I’m fixing broccoli, if I’m fixing potatoes, they’re all gonna have some type of butter in it. If I’m fixing corn, it has butter in it. So, I don’t know if this is do-able for me.*” Participants also voiced the importance of including information about spices within the model to address the taste barrier many people in the Black community have. “*There is a vaster* [sic] *range of herbs and spices out there and we have to learn to use those.*” For the adapted MIND diet model, participants voiced the importance of including foods for each of the 10 food categories that the Black community would easily recognize (i.e., green leafy vegetables—collard greens, starchy vegetable—sweet potato, and whole grains—oatmeal). Some noted ways in which they could incorporate non-preferred foods into meals. “*I don’t like certain textures, so I don’t like blueberries or berries for that reason. But I could mix them into something for the flavor.*” Others considered equivalent substitutions for personal taste or financial considerations. “*I think I like it even more if I can eat cantaloupe…berries cost a little bit more than cantaloupe and apples and things like that.*” Additionally, more specific information about serving sizes in lay language needs to be included so it is easier for people to follow. Many participants believed it was imperative to include information about the importance of water intake in relation to the MIND diet. Lastly, for each of the 10 food categories on the MIND diet, many participants indicated that they felt that it was important to include a snippet of health information (i.e., sweet potatoes contain large amounts of fiber which improves gut health and digestion [65]) so people can understand the additional benefits that food groups have for the entire body, not just benefits for brain health.

Subtheme 3—Comprehensive Education and Delivery Method: Many focus group participants voiced the importance of a dietary intervention including a faith component (i.e., scriptures, health devotionals) coupled with an ongoing education with resources (i.e., cooking classes with samples, simple recipes, preparation videos, and pocket cards). “*I know being a person of faith here at, being with my church, there are scriptures that talks [sic] about taking back what the enemy has stole [sic] from you. So that includes your mind. We often go into material things, but your mind and your health is something that it was [sic] stolen because we did not eat correctly because of lack of education and resources. So, if they are available, you know for me, myself, I would use scripture going right along with the MIND diet.*” Regarding the delivery of the education, participants across all focus groups indicated the importance of offering a hybrid learning model to engage older Black adults in an in-person setting and remotely synchronously, which would provide greater access to receiving education in the Black community. For in-person education, many participants believed it should be offered in a community academic setting such as the Alzheimer’s Disease Research Center or recreational centers.

Subtheme 4—Retention: To support retention efforts in the brain-healthy diet intervention, many participants voiced that it would be helpful to provide the community with various incentives (i.e., financial incentives such as grocery store gift cards to off-set the cost of healthy food). Since cognitive changes take time, many participants voiced the importance of highlighting short term benefits that will take place by following a healthy diet. Specifically, several participants emphasized the importance of designing an intervention that highlights the early benefits of following a healthy diet (i.e., weight loss, lower blood pressure, potential financial benefits from getting off medication), which would result in a higher retention rate in a brain-healthy diet intervention. “*Incentives might work too, if people stay on this and they know they’ve lost some weight, maybe a small token or reward, uh you know, or a trip or free meal, I don’t know, something. Incentives always help people.*”

In summary, emerged themes and subthemes from the thematic qualitative analysis increased our understanding of the quantitative results from the quantitative surveys. The quantitative data revealed the hierarchy of barriers and needs as it pertains to healthy eating while the qualitative results from the focus groups revealed specific information to “why” the barriers existed and “how” to address the barriers and needs in a culturally tailored brain-healthy diet intervention.

## 4. Discussion

The aim of this study was to describe the beliefs and attitudes that influence dietary practices among older Black adults using a community-engaged approach. More specifically, our study examined the knowledge and attitudes that influence dietary practices to inform the development of an adapted brain-healthy soul food diet intervention. Overall, our findings indicated that older Black adults perceive making healthy dietary choices as a key consideration in their overall approach to health. Prior research suggests eating habits among older adults is associated with several factors including socio-cultural attitudes, income, lower education, social isolation, and structural inequities in the neighborhood environment, which emphasizes the role social determinants have in healthy behavior [66]. In line with previous research [67], our focus groups highlighted the barriers to healthy dietary choices which were largely influenced by disparities in access, availability, and the cost of healthy foods.

This study adds to the limited research by providing context to the attitudes and beliefs shaping the current dietary practices and provides information regarding culturally specific needs related to brain-healthy dietary interventions for older Black adults. The findings indicate that adapting the MIND diet using the culturally appropriate food guidelines and practices can support buy-in by older Black adults. Previous research has indicated that Black adults associate specific foods, cooking techniques, and seasonings with their identity [14,19], which aligns with the findings from this study in which participants voiced the importance of focusing on foods aligned with the MIND diet that are well known within the Black community, as well as sharing information regarding spices and herbs to regularly incorporate within an adapted diet. Adding to the previous literature, highlighting traditional healthy soul foods and practices to an adapted MIND diet intervention is imperative in terms of having significant meaning related to passing on family traditions, which in turn provides a sense of familiarity and respect to one’s culture.

Previous research has indicated the importance of including approaches that resonate with cultural and learning styles when developing and delivering interventions [25,68,69,70]. Specifically, a holistic intervention approach that highlights the mind, body, and spirit connection has been found to be essential to support reducing health disparity risks among Black adults [69,70]. Our study aligns and adds to the current literature in which the findings from this study emphasized the importance of incorporating a holistic approach within the design of the adapted MIND diet’s educational curriculum. It is important to note that many participants in this study indicated having one of more cardiometabolic risk factors, which is consistent with previous research that has indicated that Black adults have a higher prevalence of cardiometabolic risk factors [71]. Since prevention strategies such as dietary interventions can influence brain metabolic alterations, such as gastrointestinal signaling and neurotransmitters that influences the development of neuroinflammation and neurodegenerative conditions such as AD [72,73,74,75], future studies should explore the association between dietary interventions and brain metabolism among older Black adults.

Our study has several strengths including the use of the mixed-methods approach, resulting in a more thorough knowledge imperative to inform future intervention development. Also, the use of focus groups provided a richer and more informed understanding which can add meaning to the quantitative findings. Our study was limited to a single Midwestern geographical location; therefore, it is difficult to generalize the results to a larger population. Additionally, the study primarily included participants with a bachelor’s degree or higher, which is above the average national higher education attainment for Black adults (26%) [76]. The study was also limited due to the lack of use of random sampling in which selection bias can occur with the use of purposive sampling. We were also limited by a small sample size and potential selection bias due to the fact that this project was geared towards older adults that were more technologically savvy. Several of our surveys and questionnaires have not been validated externally but were developed internally via years of iterations and discussions with the community and research experts.

## 5. Conclusions

Culturally adapted dietary interventions to improve the eating practices among older Black adults are needed to reduce the disproportionate impact AD has on the Black community. Older Black adults in our study perceived the MIND dietary model as the most feasible and acceptable to Black culture. Suggested considerations to enhance the buy-in within the Black community may include an adapted MIND dietary intervention that incorporates salient cultural characteristics and strategies, such as including a faith component within the design and delivery of the intervention. Once fully adapted, a full-scale randomized controlled trial assessing the risk reduction efficacy of a culturally tailored MIND diet will be a valuable contribution to the field.

## Figures and Tables

**Table 1 ijerph-20-06705-t001:** Participant demographics (*n* = 39) of Black older adults participating in an online survey one week before their focus group discussion.

Gender, *n* (%)	
Male	10 (25.6)
Female	29 (74.4)
Age, *n* (%)	
55–64 years	13 (33.3)
65–75 years	18 (46.2)
75 years and older	8 (20.5)
Marital Status, *n* (%)	
Single	4 (10.3)
Married	21 (53.8)
Widowed	6 (15.4)
Divorced	8 (20.5)
Education, *n* (%)	
High School	4 (10.3)
Vocational/Trade	1 (2.6)
Some College	7 (17.9)
Associate	3 (7.7)
Bachelor	9 (23.1)
Masters	13 (33.3)
Doctoral	2 (5.1)
Employment Status, *n* (%)	
Employed	11 (28.2)
Not employed	1 (2.6)
Retired	27 (69.2)
Household Income, *n* (%)	
>$25,000	2 (5.1)
$25,000–$49,999	14 (35.9)
$50,000–$74,999	13 (33.3)
$75,000–$99,999	4 (10.3)
$100,000–$124,999	3 (7.7)
$125,000–$149,999	2 (5.1)
$150,000–$199,999	1 (2.6)
Number of adults (18 and over) in household not counting yourself, *n* (%)	
0	13 (33.3)
1	18 (46.2)
2	7 (17.9)
3	0 (0)
4	1 (2.6)
Number of children under 18 in the household, *n* (%)	
0	37 (94.9)
1	0 (0)
2	2 (5.1)
Family history of dementia, *n* (%)	
Parent	5 (12.8)
Sibling	2 (5.1)
Other relative	3 (7.7)
None	29 (74.4)
Health Conditions, *n* (%)	
Hypertension	26 (66.7)
High cholesterol	20 (51.3)
Diabetes	8 (20.5)
Overweight/obesity	13 (33.3)
Religion, *n* (%)	
Christian	37 (94.9)
Other	1 (2.6)
Religiously Unaffiliated	1 (2.6)
Religious Denomination (*n* = 37), *n* (%)	
Baptist	18 (48.6)
Catholic	1 (2.7)
Methodist	8 (21.6)
Nondenominational	8 (21.6)
Pentecostal	1 (2.7)
Presbyterian	1 (2.7)

**Table 2 ijerph-20-06705-t002:** Participant responses from 39 Black older adults (*n* = 39) to eating habit questions in an online survey completed one week before their focus group discussion.

**Current Practices, *n* (%)**				
**“Do you follow a specific diet?”**				
None	19 (48.7)			
Low sodium	10 (25.6)			
Low fat	2 (5.1)			
Other **	6 (15.4)			
**“Do you practice intermittent fasting?”**				
Yes	8 (20.5)			
No	31 (79.5)			
**Meal Frequency, *n* (%)**				
**“In a typical week, how often do you eat the following meals?”**				
	0–1 Days/Week	2–3 Days/Week	4–5 Days/Week	6–7 Days/Week
Breakfast	7 (17.9)	8 (20.5)	5 (12.8)	19 (48.7)
Lunch	4 (10.3)	5 (12.8)	16 (41.0)	13 (35.9)
Dinner	1 (2.6)	1 (2.6)	4 (10.3)	33 (84.6)
**Salt Intake, *n* (%)**				
**“I add salt for cooking and preparing food at home.”**				
Very Often	5 (12.8)			
Often	6 (15.4)			
Sometimes	12 (30.8)			
Rarely	15 (38.5)			
Never	1 (2.6)			
**“I consume processed foods (not including frozen fruit or vegetables).”**				
Very Often	3 (7.7)			
Often	11 (28.2)			
Sometimes	17 (43.6)			
Rarely	7 (17.9)			
Never	1 (2.6)			
**“I think I consume too much salt.”**				
Very Often	2 (5.1)			
Often	3 (7.7)			
Sometimes	15 (38.5)			
Rarely	15 (38.5)			
Never	4 (10.3)			
**Food Preparation Practices, *n* (%)**				
**“How is your food usually prepared?” *****				
Baked	32 (82.1)			
Air fried	17 (43.6)			
Boiled	14 (35.9)			
Fried	14 (35.9)			
Grilled	18 (46.2)			
Other	5 (12.8)			

** “Other” was offered as a write-in option. Individuals responded (*n* = 6): “Mediterranean”(*n* = 1), “No red meat” (*n* = 1), “Leviticus 11”(*n* = 2), “High protein and lots of water” (*n* = 1), “Heart healthy” (*n* = 1). *** Participants could select “all that apply” so the totals will sum to over 100.

**Table 3 ijerph-20-06705-t003:** Participants indicated their interest in learning more about common health conditions, *n* (%).

	Very Interested	Interested	Somewhat Interested	Not Interested at All
Alzheimer’s Disease	19 (48.7)	16 (41.0)	3 (7.7)	1 (2.6)
Blood pressure	16 (41.0)	14 (35.9)	7 (17.9)	2 (5.1)
Cholesterol	14 (35.9)	13 (33.3)	10 (25.6)	2 (5.1)
Diabetes	14 (35.9)	15 (38.5)	7 (17.9)	3 (7.7)
Obesity	10 (25.6)	17 (43.6)	8 (20.5)	4 (10.3)

**Table 4 ijerph-20-06705-t004:** Participants identified barriers to eating a healthy diet, *n* (%).

Lack of time	6 (15.4)
Cost	8 (20.5)
Cooking Skills	5 (12.8)
Taste	12 (30.8)
Feeling Hungry	6 (15.4)
Other	16 (41)

Participants could select all that apply so the totals will sum to over 100%.

**Table 5 ijerph-20-06705-t005:** Participants indicated their preferences for nutrition education, *n* (%).

Education Component Preferences				
	Very Interested	Interested	Somewhat Interested	Not Interested at All
Meal planning and prepping	18 (46.2)	16 (41.0)	5 (12.8)	0 (0)
Eating healthy on a budget	16 (41.0)	17 (43.6)	6 (15.4)	0 (0)
Food diaries	4 (10.3)	11 (28.2)	16 (41.0)	8 (20.5)
Reading food label	6 (15.4)	22 (56.4)	10 (25.6)	1 (2.6)
Nutrition content	10 (25.6)	24 (61.5)	3 (7.7)	2 (5.1)
Water intake	17 (43.6)	20 (51.3)	2 (5.1)	0 (0)
Nutritionist or Health Education Coach Preferences				
In-person	35 (89.7)			
By phone	37 (94.9)			
Virtually	37 (94.9)			
Top 3 Location Preferences for Having Health-Related Questions Answered				
Doctor’s Office	31 (79.5)			
Website	29 (74.4)			
Social Media	12 (30.8)			

**Table 6 ijerph-20-06705-t006:** Themes, subthemes, and sample quotes from the focus group response analysis.

Main Theme	Subtheme	Sample Quotes
1. Dementia awareness	1.1 Family history 1.2 Impact on Black community/perceived cause of dementia 1.3 Role of diet in dementia	“I think that a healthy diet is important for every part of your body, especially the brain, because if we give the brain the correct nutrients, I just believe that we will, you know, function better and function longer if we eat the right things…you can’t fool the body.”
2. Practices shaping food choices and consumption	2.1 Household decision making 2.2 Food purchasing practices 2.3 Food preparation practices	“Since the pandemic, I have had….most of my shopping online and I go and pick it up, and I have a few smaller stores that I go to purchase things that um things like vegetables, things that I usually go in person to pick those out.”
3. Barriers to eating healthy	3.1 Access 3.2 Cost 3.3 Taste 3.4 Food spoilage	“Fruit and vegetables are expensive, and so whether you have, whether it’s an Aldis, Aldis is probably cheaper than a lot of places, but there’s not a lot of Aldis in some of the Black communities, it’s not a lot that is accessible.”
4. Instrumental support	4.1 Cooking education 4.2 Accessibility guidance	“It also has to be available, it’s not like we can education, we can show, we can even provide taste, but we have to make it available and show them where they can obtain or acquire these healthy foods, these recipes, these meals. If we can show them easily how to get it, or how to maintain a healthy diet, try something once or twice, but I can’t include that into my everyday lifestyle rather easily if it’s going to be difficult, it’ll be more successful if it’s easier to obtain and maintain.”
5. Elements of culturally adapted brain-healthy dietary intervention	5.1 Recruitment 5.2 Dietary model (MIND) (add seasoning/spice options, swap options, butter limitation barrier, offers variety) 5.3 Delivery method (cooking classes with samples, scriptures, recipes, collaboration w/churches) 5.4 Retention (financial incentives, emphasis on short term and long-term benefits)	“I think having a sample there or a cooking class there to show people how you can use these products, and how much they cost, and have them actually taste it, you know they can see how to utilize it. They can see that it is good and healthy for them. I think that is a good key to sort of have people, so that the change, so they can change or maybe look at things differently.”

## Data Availability

The data presented in this study are available on request from the corresponding author.

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
