# Peer review of "Development of a Culturally Adapted Dietary Intervention to Reduce Alzheimer’s Disease Risk among Older Black Adults"

_ijerph, 2023, doi:10.3390/ijerph20176705_

Round 1

Reviewer 1 Report

Some sentences are too long, some need punctuation and as a result it is sometimes hard to follow meaning. 

Reviewer 2 Report

The authors must be congratulated for their high-quality article which analyzes the importance of food consumption in older black people with AD. This is an interesting study which uses non-interventional mixed-methods to inform future intervention development and makes reference to well previous work. The manuscript is well written and easy to follow. I have some suggestions for minor revisions.

1. The addition of one paragraph to provide an overview on the importance of body cell mass index in older adults with dementia would be appropriate at the beginning, in my opinion by citing the following article: Rondanelli M, Talluri J, Peroni G, Donelli C, Guerriero F, Ferrini K, Riggi E, Sauta E, Perna S, Guido D. Beyond Body Mass Index. Is the Body Cell Mass Index (BCMI) a useful prognostic factor to describe nutritional, inflammation and muscle mass status in hospitalized elderly?: Body Cell Mass Index links in elderly. Clinical nutrition. 2018 Jun 1;37(3):934-9.

2. Perhaps the authors might consider adding one or two figures or including a graphical abstract to increase the appealability of the topic to the readers.

3. Please identify all acronyms when they first appear in the abstract. E.g., AD, MIND.

Careful re-reading by the authors is recommended to take care of some minor editorial blemishes including grammar, punctuation, spelling, space, misplaced words and improvement of overall readability.

Round 2

Reviewer 1 Report

This is a great topic, I look forward to reading about your follow-on study.